# Residents’ Behavioral Intention of Environmental Governance and Its Influencing Factors: Based on a Multidimensional Willingness Measure Perspective

**DOI:** 10.3390/ijerph192214734

**Published:** 2022-11-09

**Authors:** Shijie Li, Yan Xia, Rongbo Xiao, Haiyan Jiang

**Affiliations:** 1School of Architecture and Urban Planning, Guangdong University of Technology, Guangzhou 510090, China; 2Landscape Planning and Ecological Restoration Research Center, Guangdong University of Technology, Guangzhou 510090, China; 3Guangdong Industrial Contaminated Site Remediation Technology and Equipment Engineering Research Center, Guangdong University of Technology, Guangzhou 510006, China; 4School of Environmental Science and Engineering, Guangdong University of Technology, Guangzhou 510006, China

**Keywords:** sustainable land use, environmental governance, land pollution abatement, theory of planned behavior

## Abstract

The rapid development of industrialization has brought about a huge demand for mineral resources, and the mining industry has posed a threat to sustainable land use while promoting economic development. In the context of collaborative governance, residents are an important aspect of land pollution abatement. Therefore, understanding residents’ willingness to participate in environmental governance and exploring its influencing factors have important implications for improving the motivation of residents to participate in environmental governance and improve the local habitat. Using the multidimensional willingness measurement data of rural households’ perception of mining environment governance around the Dabaoshan mining area in Shaoguan in 2020, based on the extended theory of planned behavior, this study established a multidimensional measurement of willingness, including willingness to participate, willingness to pay, and willingness to mobilize, and used structural equation modeling to explore the factors influencing residents’ behavioral intention of environmental governance. The results suggest that behavioral attitudes had a positive effect on willingness to participate and willingness to pay. In addition, subjective norms had no significant effect on willingness to participate but were negatively related to willingness to pay. Although perceived behavioral control had no significant effect on willingness to participate, it had a positive effect on willingness to pay. In addition, the results also show that the willingness to pay and willingness to participate of farmers were positively related to their willingness to mobilize. Based on the above findings, this study proposes some policy implications to improve residents’ behavioral intention of land pollution abatement, including strengthening value perception, improving subjective awareness, building communication platforms, and improving personal capacity.

## 1. Introduction

The mining industry not only promotes rapid economic development, but also brings unprecedented environmental challenges. The mining industry, as a basic industry and one of the most critical pillar industries of China’s national economy, provides rich raw materials for industrial development. However, during mining operations, the removal of surface vegetation, the overflow of tailing wastewater, and the discharge of waste gas containing heavy metals can cause serious damage to the soil and to hydrological and atmospheric systems. This can damage animal, plant, and human ecosystems [1,2]. In addressing these problems, the Chinese government has considered mine environmental governance as one of the most important issues affecting people’s livelihoods. It is important for government departments to establish a governance system from top to bottom. The public’s behavioral intention and specific actions are the key factors in the smooth implementation of sustainable governance work [3]. However, existing studies have shown that the public’s intention to participate in mine governance is not strong, and the factors affecting the public’s behavior intention are not clear [4]. At a time when collaborative governance is advocated, it is crucial to understand and promote public participation [5,6].

In the context of studies based on the public perspective, the existing research mainly focuses on the public’s perception of the current situation of environmental pollution. This is because the public’s perception of the current situation of environmental pollution and its perception of economic, health, and lifestyle risks can reflect the current environmental and social status quo [7]. In addition, public awareness is an important factor in supporting the government’s environmental governance [8]. The research on environmental pollution perception mainly includes the following two aspects: (1) The differences in the public perception of environmental pollution risk in different regions. Pu et al.’s (2019) measurement results show significant differences in public perceptions and attitudes toward air pollution in different regions. Northeast China and some coastal areas have a higher risk perception of air pollution [9,10,11,12]. This is consistent with Li et al.’s 2021 study [10]. (2) People’s perceptions of the current situation of environmental pollution differ in different social backgrounds [13]. Withanachchi et al. (2018) concluded that people with different income sources have a significantly different cognition of pollution perception [7]. Maurice et al.’s (2019) research shows that people with different religious beliefs, cultural patterns, or ways of acquiring information have different perceptions of exposure to relevant risks [7,11,13,14]. In addition, R. Stoffle et al. found that even if two regions have similar social and cultural backgrounds, the public perception of the same project risk in the two regions is significantly different. Scholars believe that the reason for this phenomenon may be risk perception shadow (RPS), which is defined as a situation in which the public has been exposed to some form of pollution in a geographical or cultural region, and regardless of whether residents have the same political, geographic, or social environment, past pollution exposure will make the public in these areas have a similar risk perception [15].

Some scholars also studied the public behavior intention of land pollution abatement at a micro level, analyzing the influencing factors of residents’ behavior intention to participate in land pollution abatement [16,17]. Based on the theory of planned behavior, most scholars include the public’s personal psychological factors, such as attitudes, values, and personal social networks, in the influencing factors of behavioral intention. Moral norms, environmental concerns, egoism, and community attachment can influence the public’s behavioral intention [18,19,20,21,22]. Previous studies have focused on the antecedents of residents’ intention to participate in land pollution abatement [21,23] and proposed policy suggestions [23,24]. The interpretation and measurement of the behavioral intentions of the public have given rise to two main categories. One is the study of willingness to participate, mainly involving “investing in work” [21,25]. The other is the study of willingness to pay, based on “investment and payment” under macro policies and measures [26,27,28]. Few scholars have analyzed the correlation and difference between the two types of factors influencing willingness.

According to the theory of innovation diffusion, innovation is an idea, practice, or thing that is considered novel by an individual or other adoptive unit [29]. Engineering related to mine land pollution abatement can be considered an innovation that needs to be gradually accepted by the public. When a new idea is part of the first public acceptance, its influence on the perimeter of the group and on the rural residents of cohesive affinity and frequent interaction relative to the city is higher. Residents who are willing to adopt an innovation may be willing to mobilize those living near them to participate in the innovation, thereby improving the general acceptance. In the case of ecological mine restoration, when some groups have a willingness to participate in the land pollution abatement of mines or have a willingness to pay, their behavior will affect the intentions of nearby groups. If people are willing to take the initiative to influence those living nearby, they may have a significant impact, thereby helping to improve the overall behavioral willingness of local people to participate in land pollution abatement. The interest of this study is whether residents’ willingness to participate or willingness to pay has a positive impact on their willingness to mobilize others. There are only a few studies on this issue in the existing literature. Therefore, this paper proposes two questions: (1) In the theoretical model of planned behavior, what are the antecedents of willingness to participate and willingness to pay, and what are the differences between them? (2) Is there a significant relationship between residents’ willingness to participate in and pay for land pollution abatement and their willingness to mobilize others?

The Chinese government attaches great importance to the problem of mine land pollution, and a diversified governance system has gradually been formed for mine pollution control. In 2017, the Chinese government fully implemented the third-party governance model; that is, by purchasing services, the government entrusted enterprises or research institutes with strong qualifications for soil-heavy metal pollution remediation or landscape reconstruction as the third party that undertook certain governance work and gradually realized the standardization and specialization of remediation and governance of local land heavy metal pollution or landscape damage. The Dabaoshan mine is the largest polymetallic sulfide mine in South China that has a long mining history, and the current governance model adopted by the local government is a third-party governance model. Currently, many enterprises and research institutes have settled in, which is the demonstration area of the third-party governance model of mines in China (Table 1). Based on this, this study will analyze the cognitive survey data of farmers in the Dabaoshan mining area to explore the two questions raised above, with a view to providing policy guidance for the Dabaoshan mining area and similar mines.

The remainder of this paper is organized as follows: the second section presents the framework of the theory of planned behavior and the hypothesis model; the third section includes the environmental survey, questionnaire design, data acquisition and analysis methods, and the structural equation model and related parameters of the statistical results; the fourth section shows the results for the relevant factors (behavioral attitude, subjective norms, and perceived behavioral control) and the impact of the public’s willingness to participate and willingness to pay on the willingness to influence others. The significance of the results is discussed, and some policy suggestions are proposed.

## 2. Model Construction

The residential environment of the mining area has the property of a pure public good. As rational, economic, and social people, a pro-environmental behavior for mining area residents is to participate in land pollution abatement [30]. Based on previous studies, this paper explores the influencing factors. It uses the theory of planned behavior in combination with the theory of innovation diffusion, extending the research from primary intention to the mobilization of the intentions of others.

### 2.1. Theory of Planned Behavior

The theory of planned behavior was proposed by Ajzen in 1991. It is a widely used theoretical model in the field of social psychology used to study the determinants of conscious behavior. It is an extension of the theory of rational behavior. It is based on the following hypothesis: The individual is rational and can be guided to engage in a certain activity through acquired information. The subject’s willingness is the direct determinant of whether the behavior occurs, and it is linearly related to the individual’s behavioral variables. However, individual willingness is simultaneously affected by the three factors involved in behavior, namely attitude, subjective norms, and perceived behavior control. The theory of planned behavior has been widely used in many fields, such as social psychology, management, and economics, providing a theoretical basis for the analysis of individual decision-making behaviors [31,32,33,34].

### 2.2. Model Framework and Research Assumptions

The theory of planned behavior assumes that the individual’s behavior intention is determined by the three hypothetical concepts of behavioral attitude, subjective norms, and perceived behavioral control. Behavioral attitude refers to the individual’s positive or negative evaluation of the behavior involved, which mainly depends on moral obligations and personal preferences. Previous studies have shown that positive behavioral attitudes will have a positive impact on the public’s willingness to act [24,35,36]. Wang’s research shows that tourists’ responsible environmental behavior in tourism activities is positively influenced by their attitude toward environmental behavior [37]. Subjective norms refer to the social pressure that an individual feels when performing or not performing a certain behavior [38]. Most studies use subjective norms to reflect the attitudes or social pressures of the surrounding people as perceived by individuals [32,34]. Previous studies have confirmed that the greater the influence of important people such as family and friends, the more likely the formation of behaviors that are beneficial to the environment [3,19]. Perceived behavioral control refers to the perceived difficulty of an individual’s behavior, reflecting past experiences and perceived barriers [38]. Perceived behavioral control mainly measures the cost and benefit of an individual’s performance of a behavior, such as the time, money, and energy spent [3]. Existing research shows that individuals will have a willingness to participate in easy behaviors [39,40]. Based on the above analysis, the paper proposes six hypotheses:

**Hypothesis** **1 (H1).***Residents’ behavioral attitudes are significantly related to their willingness to participate in land pollution abatement*.

**Hypothesis** **2 (H2).**
*Residents’ behavioral attitudes are significantly related to their willingness to pay for land pollution abatement.*


**Hypothesis** **3 (H3).**
*Residents’ subjective norms are significantly related to their willingness to participate in land pollution abatement.*


**Hypothesis** **4 (H4).**
*Residents’ subjective norms are significantly related to their willingness to pay for land pollution abatement.*


**Hypothesis** **5 (H5).**
*Residents’ perceived behavioral control is significantly related to their willingness to participate in land pollution abatement.*


**Hypothesis** **6 (H6).**
*Residents’ perceived behavioral control is significantly related to their willingness to pay for land pollution abatement.*


In addition, to explore the relationship between the willingness of local residents to participate and pay and their willingness to mobilize, we propose the following two hypotheses:

**Hypothesis** **7 (H7).**
*Residents’ willingness to participate in land pollution abatement is significantly related to their willingness to mobilize.*


**Hypothesis** **8 (H8).**
*Residents’ willingness to pay for land pollution abatement is significantly related to their willingness to mobilize.*


Based on the above assumptions, this study constructed a multidimensional willingness measurement system model, mainly divided into two levels: the first level explores the influencing factors of residents’ willingness to participate and willingness to pay, and the second level explores the impact of residents’ willingness to participate and willingness to pay on their willingness to mobilize. Figure 1 show the model framework.

## 3. Methods

### 3.1. Study Area

As Figure 2 shows, the Dabaoshan mining area is located in Shaoguan City, Guangdong Province, China, spanning Qujiang and Wengyuan Counties. The mining area is about 578.96 hm^2^ and is located in a warm, humid, and rainy area. Construction work on the Dabaoshan mining area began in 1958, and subsequent illegal civilian mining activities have become increasingly problematic. The 18th National Congress of the Communist Party of China proposed for the first time building a beautiful China and prioritized ecological improvement initiatives. The environmental issues in Dabaoshan have been a focus of attention. The central government and local governments have issued a series of policy documents requiring the treatment of mine land pollution. In the last decade, the government has conducted a series of works for land pollution control (Figure 3). In 2012, the civilian mining area was closed. Although illegal mining in the surrounding area was controlled, acidic water and heavy metal pollution remained. In 2013, the relevant departments of Guangdong Province began to remediate the mining area and the surrounding environment, and the LiWu external drainage expansion project was officially completed in 2015 (Figure 4). Subsequently, the sewage treatment capacity in the Dabaoshan area was further improved. After the 2016 release of the “Ten Articles” in the Action Plan for Soil Pollution Prevention and Control, the Dabaoshan mining area became a leading area for pollution prevention and control. At the end of 2016, sewage treatment was established. After that, the heavy metal content in the irrigation water of LiangQiao and Shangba Village dropped from 0.02 to 4.50 mg·L^−1^ to below the detection limit (Table 2) [41,42]. Soon after, the Institute of Subtropical Agricultural Ecology, the Chinese Academy of Sciences, the South China Agricultural University, Guangzhou Caomufan Environmental Technology Co., Ltd., and other scientific research institutes, universities, and enterprises engaged and cooperated with local governments to carry out research projects on pollution and practical work for pollution prevention and land pollution abatement (Table 1). After nearly 10 years of governance, the ecological environment of the Dabaoshan mining area has been significantly improved

### 3.2. Data Collection

The research team conducted field investigations in the Dabaoshan mining area from November to December 2019 to understand the scope of pollution impacts in the mining area, the land pollution abatement process, and the actual effects of governance. The survey was conducted from January to June 2020. Then, the research team carried out an investigation of the background research literature and engaged in methodological discussions. The theories of planned behavior and structural equation modeling methods were identified and selected. From July to December 2020, the design of the “Dabaoshan Residents’ willingness for land pollution abatement survey” questionnaire was completed, and the implementation plan of the questionnaire survey was formulated. The plan included the sample selection, the pre-survey, and the formal survey. The questionnaire consists of two parts: one is to collect personal and family information, and the other is to investigate the awareness of and willingness to participate in land pollution abatement. Personal and family information was supplied according to the actual situation. Behavioral willingness was measured on a Likert scale from 1 to 5, with 1 to 5 being very unwilling, unwilling, average, willing, and very willing. The cognitive variables were also measured on a 5-point likert scale. After the questionnaire design was completed in December 2020, a preliminary survey was conducted in Tangxin Village and Fupi Village in January 2021 under the advice of local environmental restoration experts to verify the readability of the questionnaire, the rationality of the question setting, and the authenticity of the answers. This was followed by large-scale questionnaire distribution in March and April 2021. The questionnaire was distributed using a simple random sampling method, with the Dabaoshan mining site as the center and a radius of 10 km as the survey area. According to the risk perception mapping sampling procedure designed by R. Stoffle et al., it is thought that the direction and distance relative to the study area are parameters affecting RPS, which is of reference significance for this study. Therefore, considering the relatively uneven distribution of villages and the trend of polluted rivers [15], this study finally took topography and hydrology factors into consideration, and the questionnaire area was divided into 10 circles at 1 km intervals. An administrative village was selected as the sampling site within each circle. Taking into consideration instances of village relocation or insufficient population, the 10 administrative villages in the figure were selected as the questionnaire survey objects (Figure 5). Approximately 50 households were selected from each village as the interviewees; only one family member from each household was allowed to participate in the questionnaire. Most of the interviewees are permanent residents. As such, they may have a better understanding of the local environment and governance situation, thereby ensuring the reliability of the survey data. In this survey, a total of 600 questionnaires were collected, 540 of which were valid (Table 3).

### 3.3. Variable Description

Table 4 shows descriptive statistics for the variables in this study. Residents’ behavior and attitude are measured by individual cognition of social, economic, and ecological values of land pollution abatement. In general, subjective norms are reflected through the perception of attitudes of family members and friends. However, as the leaders of land pollution abatement, relevant measures of the government and mines can provide a better social atmosphere for governance and, to some extent, generate social pressure. Therefore, in this study, the perception of the attitude of the government and mining enterprises was added to reflect subjective norms to measure the impact of government and mining measures on the intention of mine residents to participate in land pollution abatement. Perceived behavioral control is reflected by the individual’s perceived energy and ability and the influence of their participation in governance.

### 3.4. Descriptive Analysis

Table 5 shows the basic characteristics of the interviewed farmers. Among the 540 interviewees, in terms of personal characteristics, 59.81% of the respondents were male. The age of the respondents was mainly under 50, accounting for 75.37% of the total sample. The level of education was generally low, with junior high school and below accounting for 73.70% of the respondents; occupations were mainly concentrated in farming and part-time work, accounting for 70.93% of the total number of samples.

Table 3 shows that residents who live close to the mine have a higher willingness to participate. As the distance increases, the willingness of residents to participate decreases. This may be due to the more serious pollution in the villages closer to the mine. The greater the changes before and after environmental treatment, the higher the residents’ recognition of environmental treatment and the stronger their willingness to participate. The public’s willingness to participate in land pollution abatement-related work is relatively positive. A total of 79% of the respondents expressed their willingness to participate in land pollution abatement work. However, Figure 6 shows that the public’s willingness to pay for land pollution abatement is not high. Nearly half of all respondents were unwilling to pay for land pollution abatement. Only 284 of the 425 respondents willing to participate were willing to pay for land pollution abatement, accounting for only 66.80%. Figure 7 shows that willingness to participate and willingness to pay have different effects on mobilization willingness.

### 3.5. Measurement Model

The structural equation model (SEM) measures unobservable variables by observational variables and uses various indicators to measure the fitting degree of the model [43]. Therefore, this study adopted the SEM, and used convergence validity, variable synthesis reliability, discriminant validity, and content validity to test the validity of this study. Table 6 shows the results of the convergent validity, which indicate the variable integrated reliability under the same dimension. In this study, all factor loadings were greater than 0.5, and the average variances extracted were greater than 0.5, showing that the questionnaire has good convergent validity. The variable integrated reliability is greater than 0.6. This shows that the model is of good quality [44]. Content validity refers to the extent to which the content of the questionnaire is appropriate and consistent. This paper referred to previous studies; therefore, it has content validity. The square root of the AVE of the latent variables was compared with the correlation coefficient between the latent variables in order to test the discriminant validity. As shown in Table 7, the square roots of the latent variables were all higher than the correlation coefficients between them and other latent variables. Therefore, the discriminant validity is verified.

### 3.6. Structural Model

This study used SEM based on covariance to analyze the influence of each factor. After establishing the model, the degree of influence on the willingness to participate and willingness to pay was determined according to the significance of the path coefficient, and the rationality of the model was verified by the test index. The inspection criteria and results are shown in Table 8 [45,46]. Table 9 shows the results of the model; the model was implemented using AMOS 23.0.

## 4. Results and Policy Implications

After processing the data, we determined the influencing factors of residents’ willingness to participate in land pollution abatement and their willingness to pay (Figure 8; Table 9). We then proposed some targeted policy recommendations.

### 4.1. Results

(1)Attitudes have a positive impact on residents’ willingness to participate and willingness to pay.

The results of our analysis show that attitudes are significantly positively correlated with residents’ willingness to participate and willingness to pay; the impact factors were 0.88 and 0.37, respectively. Therefore, Hypothesis 1 and Hypothesis 2 are valid. When residents realize that land pollution abatement can bring social value, increase their income, and improve their living environment, they are more inclined to participate in land pollution abatement-related activities and pay the related expenses. However, the influence of behavioral attitude on residents’ willingness to pay is higher than on their willingness to participate. This may be because willingness to pay in ecological economics refers to the beneficiary’s intention and desire to pay for public goods that have no market price for their own benefit. This implies that residents must feel that land pollution abatement is beneficial to their lives before they become more willing to pay. However, willingness to participate is different. It is not necessary to experience an in-depth perception of the potential positive changes brought about by land pollution abatement, but there may be related participation behaviors. Regardless of whether land pollution abatement can bring social value or improve quality, residents can experience potential positive changes brought about by land pollution abatement. Therefore, the above factors may have a higher impact on willingness to pay than on willingness to participate.

(2)Subjective norms do not affect residents’ willingness to participate, but significantly affect their willingness to pay and are negatively correlated.

The impact factor of subjective norms on local residents’ willingness to participate was −0.04, and the result was not significant; that is, subjective norms do not affect local residents’ willingness to participate. Hypothesis 3 is not valid, in line with some existing research results [3,21]. According to the actual situation of the study area, some governance behaviors require residents to participate actively. They may be asked to participate in experimental field experiments on their own land in paid vegetation restoration activities. Residents may choose to assist, and some residents may regard these activities as a source of income. Sufficient consideration may not have been given to the policy impact of the government’s and mining companies’ actions on land pollution abatement and to the attitudes of important people such as neighbors and friends. The small number of local residents involved in land pollution abatement activities could be because group social pressure has yet to be formed. The impact factor of subjective norms on residents’ willingness to pay was −0.64, showing a significant negative correlation, supporting Hypothesis 4. The results show that the more relevant the environmental policies, and the more inclined important people around are to pay for land pollution abatement, the less willing residents are to pay for land pollution abatement. That is to say, when others are willing to pay for land pollution abatement for their own benefit, and when local residents do not have to pay fees to enjoy the results, they may not choose to “pay the bill.” Second, there may be some residents who fail to realize that land pollution abatement is an activity with the public as the main body. Environmental sustainability cannot be achieved solely through the efforts of the government and mining enterprises. The more that residents feel that the government and mining companies are actively leading land pollution abatement-related work, the more their willingness to pay is reduced.

(3)Perceived behavioral control has a significant influence on residents’ willingness to pay.

Perceived behavioral control does not affect the willingness of residents to participate, with an impact factor of only 0.23, making Hypothesis 5 invalid; however, the impact factor of their willingness to pay was 0.85, a relatively large impact, making Hypothesis 6 valid. When residents think that they have enough energy and ability, and that the payment of expenses will not impact too much on their lives, they are more willing to pay related expenses for land pollution abatement. Compared with willingness to participate, willingness to pay is a stronger route of participation, with higher requirements on the economic status and ability of residents. According to Maslow’s demand theory, residents’ willingness to pay for environmental improvement is equivalent to their need for self-realization. Their willingness to participate may be the result of necessity. The influencing factors will be different when the two needs are at different levels.

(4)Willingness to participate and willingness to pay can significantly affect willingness to mobilize, and the impact factor of willingness to participate is higher.

Residents’ willingness to participate and willingness to pay are significantly positively correlated with their willingness to mobilize others; the impact factors were 0.71 and 0.34, respectively. Hypothesis 7 and Hypothesis 8 are both valid, and the willingness to participate has a greater impact on the willingness to mobilize. This shows that when individuals have a willingness to participate or pay, they are willing to mobilize the important people around them to participate in the land pollution abatement of the mine, thereby expanding the social network of residents who are willing to participate or willing to pay or increase their social influence. It is helpful to promote the overall willingness of local residents to participate or pay. Residents’ willingness to participate has a greater impact on their willingness to mobilize others than their willingness to pay. This may be because residents living near the mine have a deeper understanding of the advantages of land pollution abatement after personally participating in land pollution abatement-related work. They become more willing to encourage others to participate in activities related to land pollution abatement. Groups who are willing to pay will spend money on land pollution abatement. They enjoy the benefits of land pollution abatement, such as cleaner air, a beautiful environment, and healthier communities. The changes brought about as a result of land pollution abatement make it easier to mobilize others to participate in land pollution abatement.

### 4.2. Policy Implications

(1)Recognition: strengthen the social, economic, and ecological value perception of land pollution abatement.

Generally speaking, the economic interests of local residents are extremely important for long-term sustainable governance plans [47]. Interest factors restrict residents’ willingness to act. The results of this study show that local residents’ recognition of the results of the land pollution abatement of mining has played an important role in influencing their behavior and attitudes, and that local residents are concerned about the specific benefits of land pollution abatement rather than the overall situation.

Therefore, when the government promotes the advantages of land pollution abatement, it should not simply make residents understand the relevant policies and environmental benefits. These macro concepts are not effective. Instead, the government should show real cases of land pollution abatement in exhibitions or visits to experimental sites and other related activities, whereby residents can appreciate the economic value and social significance that land pollution abatement policies can bring, for example, though exhibitions or bulletin boards to compare the content of heavy metals in the soil, green space coverage, or crop yield before and after the management, and by using these indicators closely related to the lives of farmers to quantitatively evaluate the benefits of the management work. Secondly, it is also necessary to use modern information technology to deliver simple and easy-to-understand knowledge to local residents, promote articles on WeChat and QQ, and use short promotional videos on Douyin, Kuaishou, and Toutiao. The benefits of land pollution abatement that are closely related to life are easier to perceive and understand. This understanding enhances residents’ environmental awareness, eliminates the “free-ride” psychology, and comprehensively strengthens residents’ awareness of the meaning of land pollution abatement.

(2)Empowerment: cultivate residents’ subjective consciousness.

Although the government is the leader of land pollution abatement, residents are undoubtedly the driving force. The effectiveness and sustainability of land pollution abatement still depend on local residents [48]. However, the current governance model is mainly a government-led “top-down” model. In most cases, the local government of Dabaoshan directly negotiates with the governance enterprises or scientific research institutes. The participation of local farmers in this process is very low, and there are few opportunities for participation. In order to succeed in the governance model, it is far from enough to establish a relationship between the government and enterprises or scientific research institutions. Generally speaking, the third-party governance enterprises or scientific research institutions need the cooperation of village committees, rural enterprises, or village collective economic organizations to receive support from local farmers; therefore, the government should also promote communication and exchange between local farmers and enterprises or research institutes. Therefore, first, an information exchange platform should be built between the government and local residents. When people can access information on government and governance-related work, they can better understand the working process of environmental control and the problems in governance. However, to eliminate the information asymmetry between the government and residents, the government or the mining company should take responsibility. Communication between the government and residents should be strengthened to cultivate public awareness. Second, in decision-making related to land pollution abatement, residents should have the right to participate and vote. Therefore, residents should be fully guided to participate in decision-making activities on related matters, so that they can express their demands and opinions, thereby enhancing their sense of participation and stimulating their motivation to participate.

(3)Cooperation: build a neighborhood communication platform.

When making decisions, it is difficult for an individual not to consider the social relationship between their actions and those of others. They will not be independent of social and cultural influences [49]. Their attitudes are easily influenced when they perceive that people around them are participating in an activity. The characteristics of the “acquaintance society” in rural areas can enable land pollution abatement to build an identity mechanism with relationship networks and trust mechanisms, because residents’ attitudes are easily influenced when they perceive that people around them are participating in an activity. In particular, respected persons, elites, party members, and cadres in the countryside can use their exemplary role to drive the recognition of the surrounding people. The government could build a public exchange platform for land pollution abatement information on mining, regularly holding demonstration learning activities and recommending advanced land pollution abatement models. This will enable understanding of the attitudes and behaviors of the surrounding people and will increase social pressure. Since the willingness to participate and the willingness to pay have a significant impact on the willingness to mobilize others, those willing to participate could serve as examples encouraging their friends and relatives to participate in land pollution abatement.

(4)Improvement: enhance the ability to participate.

Residents’ behavioral intentions are inseparable from their abilities and energy. It is vital to explore new ways to improve residents’ abilities by promoting relevant training, guidance, and vocational education. Many scientific research institutes, universities, and local units, such as the Institute of Botany of the Chinese Academy of Sciences, the South China University of Technology, the South China Agricultural University, the Guangdong Academy of Environmental Sciences, and the Shaoguan Science and Technology Bureau, are currently conducting experimentation and exploration in Dabaoshan. At this time, the government can organize relevant scholars and technicians to conduct on-site technical training for local residents. In addition, some agricultural assistance projects could expand residents’ employment possibilities, change their traditional employment methods, increase their income, and enhance their sense of self-efficacy, enabling them to make scientific and reasonable decisions.

## 5. Conclusions

Based on the extended theory of planned behavior, this study established a multi-dimensional willingness measure including willingness to participate, willingness to pay, and willingness to mobilize. Survey data of farmers around the Dabaoshan mining area in Shaoguan in 2020 were used for empirical research, and the structural equation model was used. The factors that influence residents’ willingness to conduct environmental governance were discussed.

The following conclusions can be drawn. Behavioral attitude will affect farmers’ willingness to participate and willingness to pay. Subjective norms do not affect farmers’ willingness to participate. It is worth noting that subjective norms are negatively correlated with willingness to pay, indicating that, due to the particularities of environmental governance, some residents may have a “free-rider” mentality. Perceived behavioral control has a positive impact on farmers’ willingness to pay, which shows that in the process of environmental governance, willingness to pay can better reflect the strong participation of farmers. The results show that willingness to pay and willingness to participate both have an impact on farmers’ willingness to mobilize. Based on the above findings, this study proposed some policy initiatives to improve residents’ behavioral intention of land pollution abatement, including strengthening value perception, improving subjective awareness, building communication platforms, and improving personal capacity.

The study site is only a single mining area in Dabaoshan. However, the Dabaoshan mining area is a representative model mining area in China where third-party mining subjects participate in the implementation of governance measures. Nonetheless, the research conclusions and policy recommendations are highly contextualized. To make the research findings and policy recommendations more widely applicable, they need to be applied to other areas with the same type of comprehensive analysis of mining land pollution abatement. This is one of the limitations of this study. Secondly, the Dabaoshan mining area has carried out environmental governance work for decades; therefore, policy intervention may affect the measurement results of farmers’ multidimensional willingness. We considered this bias in the process of setting the questionnaire, and thus the perceived government attitude and perceived mining enterprise attitude indicators were included in the measurement system in this study. However, our indicators are not comprehensive, which is also one of the limitations of this study.

## Figures and Tables

**Figure 1 ijerph-19-14734-f001:**
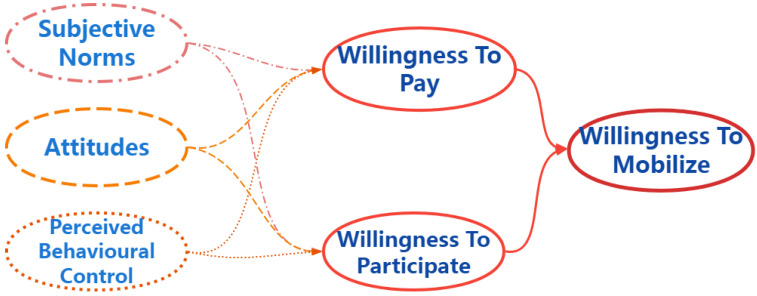
Theoretical model framework.

**Figure 2 ijerph-19-14734-f002:**
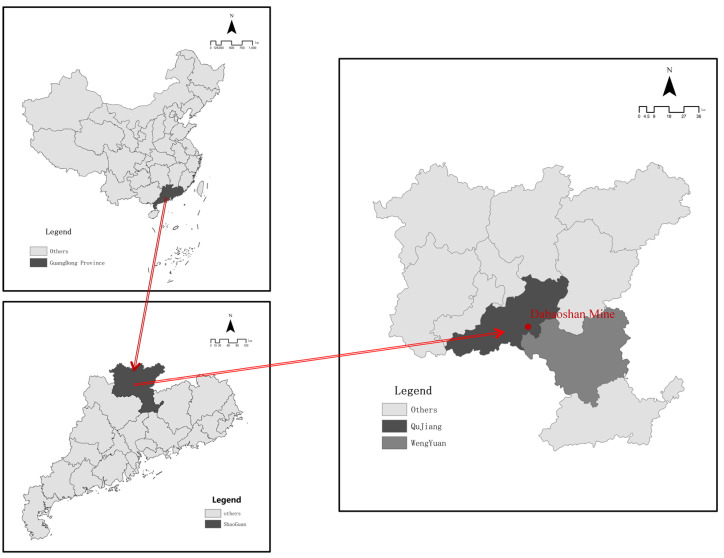
Location of the Dabaoshan mine.

**Figure 3 ijerph-19-14734-f003:**
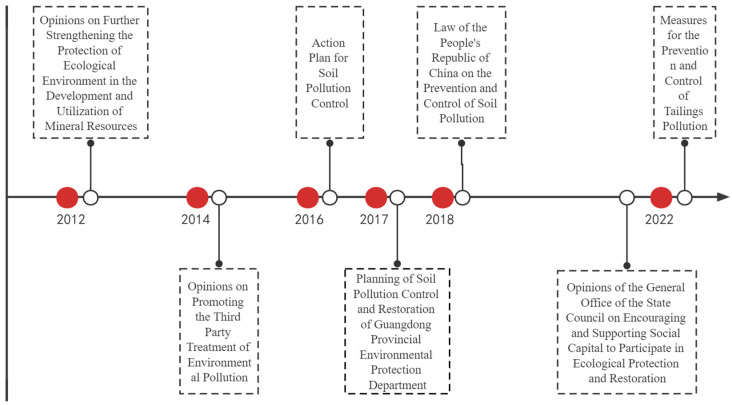
Policy documents on mine control issued by the Chinese government.

**Figure 4 ijerph-19-14734-f004:**
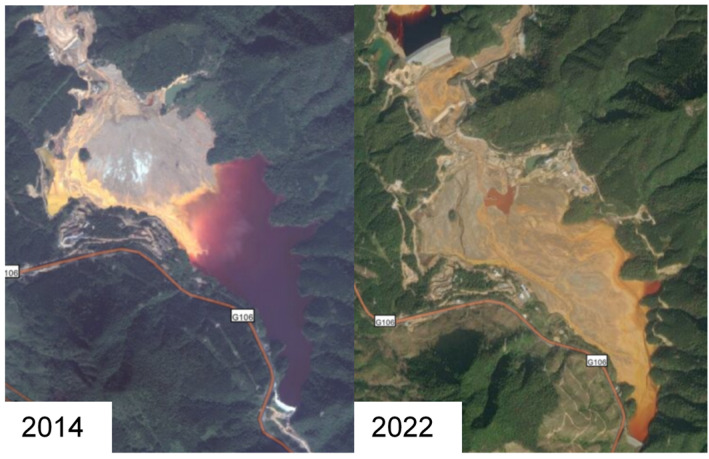
Comparison before and after the dredging project of the Liwu mud bank.

**Figure 5 ijerph-19-14734-f005:**
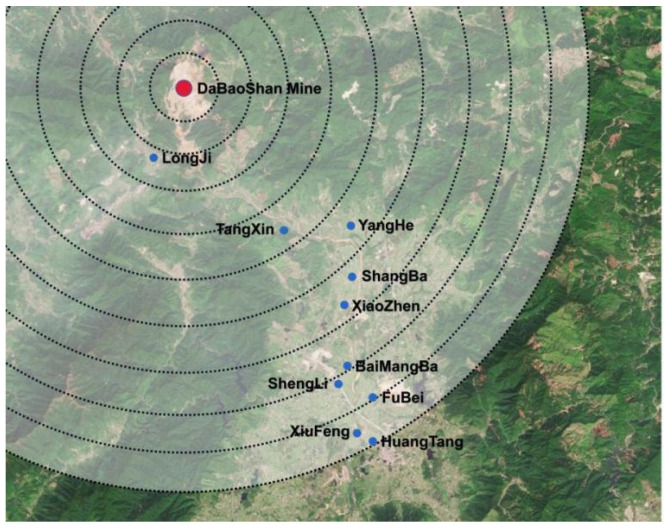
Sampling location of questionnaires.

**Figure 6 ijerph-19-14734-f006:**
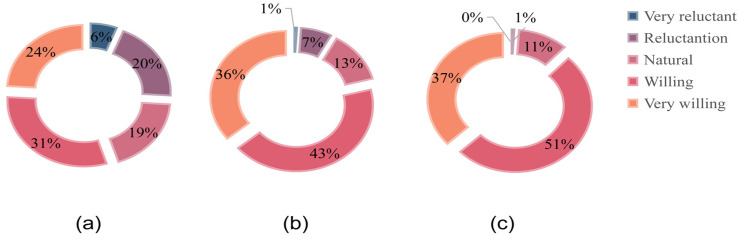
Scale diagram of respondents’ behavioral intention (**a**) willingness to participate; (**b**) willingness to pay; (**c**) willingness to mobilize.

**Figure 7 ijerph-19-14734-f007:**
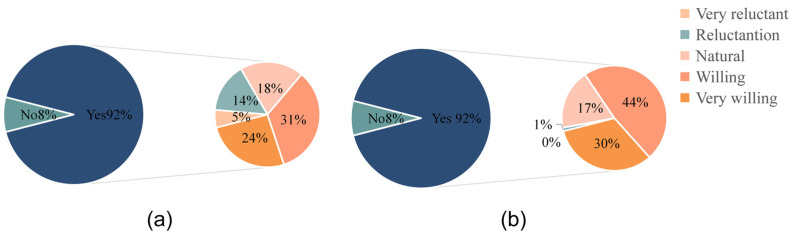
The distribution of the mobilization intention of the respondents who have the intention to participate (**a**) willingness to participate; (**b**) willingness to pay.

**Figure 8 ijerph-19-14734-f008:**
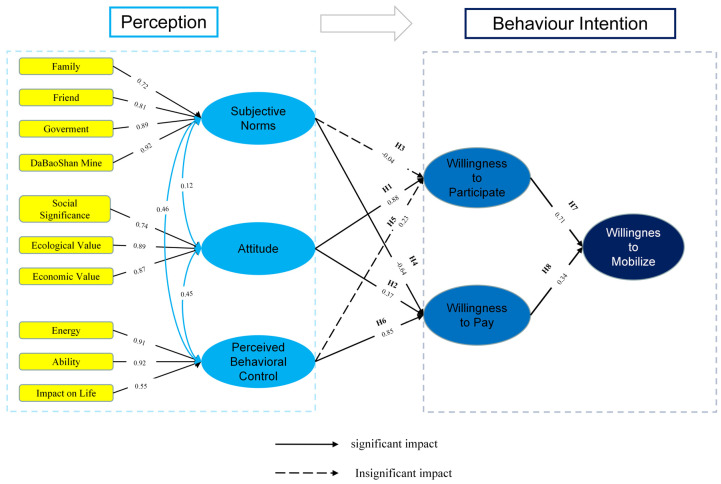
Path coefficients and model test results.

**Table 1 ijerph-19-14734-t001:** Research institutions and projects in the Dabaoshan mining area.

Name of Organization	Project Name
Grass and Wood Demonstration Base	
South China University of Technology South China Agricultural University Wuhan Zhongdi Jindun Environmental Technology Co., Ltd.	Research and development of geochemical engineering remediation technology for heavy metal pollution in farmland
Environmental Protection Research and Monitoring Institute, Ministry of Agriculture	Intercropping of maize with low accumulation of heavy metals and hyperaccumulation plants for remediation technology demonstration
Shaoguan Environmental Protection Bureau Shaoguan Agriculture Bureau Shaoguan Science and Technology Bureau Guangdong Province Environmental Science Research Institute	Farmland soil remediation verification and evaluation platform in northern Guangdong
Zhejiang Academy of Agricultural Sciences	Enhanced repair techniques of hybrid rice intercropping system with super/high enrichment plants under different rice-ripening systems
Institute of Botany Chinese Academy of Sciences	Study on selection and evaluation of maize varieties with low heavy metal accumulation
Institute of Geographic Sciences and Natural Resources Research, CAS	Intercropping-enhanced remediation techniques of rice/maize with low heavy metal accumulation and hyperenrichment plants under different conditioning modes
Guangdong Academy of Environmental Sciences	Tielong Forest Farm farmland soil planting structure adjustment test base
Guangdong Academy of Environmental Sciences	Remediation demonstration project of heavy metal contaminated soil in Tielong Forest Farm, Wengyuan County, Shaoguan City

**Table 2 ijerph-19-14734-t002:** After the establishment of the sewage treatment plant, the inlet and outlet water quality was tested [41,42].

Time	January 2016		November 2016		May 2020	
Sampling Location	LiangQiao	ShangBa	LiangQiao	ShangBa	LiangQiao	ShangBa
pH	3.86	4.78	8.7	7.9	6.12	6.68
Fe (mg∙L^−1^)	25	12	4	—	—	—
Mn (mg∙L^−1^)	15	9	—	—	—	—
Zn (mg∙L^−1^)	6	5	—	—	—	—
Cu (mg∙L^−1^)	2	1	—	—	—	—
Cd (mg∙L^−1^)	—	—	—	—	—	—
Pb (mg∙L^−1^)	0.08	—	—	—	—	—
SO_4_^2−^ (mg∙L^−1^)	1030	798	382	130	299	107

Note: — indicates below the detection limit.

**Table 3 ijerph-19-14734-t003:** Questionnaire sampling points and number of questionnaires.

Place	N	Distance	Mean (Willingness to Participate)	Mean (Willingness to Pay)
LongJi	48	2	4.81	3.54
Tangxin	40	4	4.56	4.03
Yanghe	100	5	4.37	4.23
Shangba	48	6	3.98	3.71
Xiaozhen	42	6	4.12	3.95
BaiMangBa	52	7	3.56	3.00
Shengli	58	8	3.93	2.48
Fubei	50	8	3.00	2.92
Xiufeng	50	9	3.96	3.18
Huangtang	52	9	4.02	3.13

**Table 4 ijerph-19-14734-t004:** Constructs and indicators of the theory of planned behavior model.

Construct	Indicators		Response Scale (1–5)	References Used
Attitudes	ATT1	Do you think it is meaningful for residents to participate in the land pollution abatement of the Dabaoshan mining area?	Meaningless–Very meaningful	[24]
	ATT2	Participating in the land pollution abatement of the Dabaoshan mining area can improve crop quality	Strongly disagree–Strongly agree	
	ATT3	Participating in the land pollution abatement of the Dabaoshan mining area can improve the surrounding environment	Strongly disagree–Strongly agree	
Subjective Norms	SN1	Will your family’s attitude affect your enthusiasm to participate in the land pollution abatement of the Dabaoshan mining area?	No effect–Very influential	
	SN2	Will the attitude of your neighbors and friends affect your enthusiasm to participate in the land pollution abatement of the Dabaoshan mining area?	No effect–Very influential	
	SN3	Will the attitude of the local government affect your enthusiasm to participate in the land pollution abatement of the Dabaoshan mining area?	No effect–Very influential	
	SN4	Will the attitude of the Dabaoshan Mining Company affect your enthusiasm to participate in the land pollution abatement of the Dabaoshan mining area?	No effect–Very influential	
Perceived Behavioral Control	PBC1	Do you have the energy to participate in the land pollution abatement of the Dabaoshan mining area?	No energy–very energetic	
	PBC2	Are you capable of participating in the land pollution abatement of the Dabaoshan mining area?	Incapable–very capable	
	PBC3	Participation in the Dabaoshan Mine land pollution abatement will take up your daily rest time, thereby reducing your income. (Do you agree with this statement?)	Disagree–agree strongly	
Willingness to Participate	PW1	In order to protect the ecological environment, are you willing to participate in the land pollution abatement of the Dabaoshan mining area?	Very unwilling–Very willing	
Willingness to pay	WTP1	In order to protect the ecological environment, are you willing to pay a fee for the land pollution abatement of the Dabaoshan mining area?	Very unwilling–Very willing	[21]
Willingness to mobilize	MW1	In order to protect the ecological environment, are you willing to mobilize the surrounding people to participate in the land pollution abatement of the Dabaoshan mining area?	Very unwilling–Very willing	

**Table 5 ijerph-19-14734-t005:** Demographic variables of the study participants.

Feature	Number of Samples	Proportion (%)	Feature	Number of Samples	Proportion (%)
Gender			Land contracted by the family (m^2^)		
Male	323	59.81	≤2667	291	53.89
Female	217	40.19	2667–4000	116	2148
Age			4000–5333	80	14.81
≤40	220	40.74	>5333	53	9.81
40–50	187	34.63	Annual income		
50–60	99	18.33	≤30,000	82	15.19
60–70	30	5.56	30,000–60,000	196	36.30
>70	4	0.74	60,000–90,000	111	20.56
Education			90,000–120,000	119	22.04
Illiterate or semi-literate	10	1.85	More than 120,000	32	5.93
Elementary school	132	24.44	Occupation		
Junior high school	256	47.41	Farmer	271	40.19
Senior high school	107	19.81	Worker	166	30.74
Junior college and above	35	6.48	Businessperson	52	9.63
Used to be a village cadre			Teacher	4	0.74
Yes	31	5.74	Governmental personnel	4	0.74
No	509	94.26	Student	7	1.30
			Others	36	6.67

**Table 6 ijerph-19-14734-t006:** Measurement model.

Constructs	Indicator	Factor Loadings	Average Variance Extracted (AVE)	Composite Reliability (CR)
Attitude	ATT1	0.74	0.711	0.880
	ATT2	0.90		
	ATT3	0.88		
Subjective norms	SN1	0.77	0.721	0.911
	SN2	0.85		
	SN3	0.88		
	SN4	0.89		
Perceived behavioral control	PBC1	0.89	0.664	0.851
	PBC2	0.93		
	PBC3	0.58		

**Table 7 ijerph-19-14734-t007:** Discriminant validity test.

Latent Variable	ATT	SN	PBC
ATT	0.843		
SN	0.077	0.849	
PBC	0.467 ***	0.490 ***	0.815

Note: The diagonal is the square root of the AVE; the correlations between the latent variables are underneath the diagonal. *** *p* < 0.001 level.

**Table 8 ijerph-19-14734-t008:** Criteria of test indicators for the structural equation model.

	Indicator		Criteria	Result
Absolute fit measures	PCMIN/DF		<3.00	2.837
	RMESA	Root Mean Square Error of Approximation	<0.08	0.058
	AGFI	Adjusted Goodness of Fit Index	>0.80	0.929
	GFI	Goodness of Fit Index	>0.80	0.963
	SRMR	Standard Root Mean Square Residual	<0.08	0.034
Incremental fit measures	NFI	Normed Fit Index	>0.90	0.971
	CFI	Comparative Fit Index	>0.90	0.981
	IFI	Incremental fit index	>0.90	0.981
	TLI	Tucker Lewis Index	>0.90	0.969
Parsimonious fit measures	PGFI	Parsimonious Goodness of Fit Index	>0.50	0.508
	PNFI	Parsimonious Normed Fit Index	>0.50	0.598
	PCFI	Parsimonious Comparative Fit Index	>0.50	0.604

**Table 9 ijerph-19-14734-t009:** Structural equation modeling results: standardized direct, indirect, and total effects.

	Endogenous Variables		
	Willingness to Participate	Willingness to Pay	Motivation Intention
Direct effects			
Attitude	0.883 ***	0.372 ***	
Subjective norms	−0.036	−0.639 ***	
Perceived behavioral control	0.229 ***	0.848 ***	
Willingness to participate			0.340 ***
Willingness to pay			0.714 ***
Indirect effects			
Attitude			0.757 ***
Subjective norms			−0.242 ***
Perceived Behavioral control			0.452 ***
Willingness to participate			
Willingness to pay			
Total effects			
Attitude	0.883 ***	0.372 ***	0.757 ***
Subjective norms	−0.036 ***	0.639 ***	−0.242 ***
Perceived behavioral control	0.229 ***	0.848 ***	0.452 ***
Willingness to participate			0.741 ***
Willingness to pay			0.340 ***

Note: *** *p* < 0.001 level.

## Data Availability

Not applicable.

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
