# Peer review of "Residents’ Behavioral Intention of Environmental Governance and Its Influencing Factors: Based on a Multidimensional Willingness Measure Perspective"

_ijerph, 2022, doi:10.3390/ijerph192214734_

Round 1

Reviewer 1 Report

This is a standard application of willingness to support or act regarding a specific project or mitigation of project. The work is important and should be published. Well done research.

The notion that people respond in terms of distance to study target area is not supported by other research. See Risk Perception Mappings research, where the people at risk were influenced by their perception of how the threat could impact them. So the potential placement of a radioactive waste facility was increased by the flow of water. So downstream people perceived higher risk and had different responses to the proposal. See R. Stoffle (et. al) 1991 Risk Perception Mapping article (American Anthropologist)which was the first to identify this risk and response variable. There is a rather large number of at risk studies today.

It remains a good paper and should be published.

Reviewer 2 Report

This study is focused on understanding the willingness of citizens in participating in land pollution abatement while taking into account influencing factors, based on an extended theory of planned behaviour. Its site is situated in Dabaoshan mining area, Ghangzhao, China. While the methodology is well designed and carried out, my first objection to this work is that it does not mention the study site in the Abstract and Introduction, making the results look too general. The authors must state the study site and its limitations in these parts of the paper and also speak about the limits in generalizabilty, as the study is highly contextualized. The second objection is that the authors briefly mention the policy without providing details of the various abatements taken to reduce land pollution over a decade. These should be detailed for readers to understand better the quality of the policy, and the type of interventions used. In this regard, Figures 3a and 3b are insufficient to demonstrate improvements due to the policy. The authors must present quantitative, scientific evidence, such as reduction in pollution levels, etc., to demonstrate the improvement along with references, otherwise these are largely unfounded.  In this regard, authors must also account as to how their model accounts for policy intervention and its efficacy influencing their results and the limitations of the study, as it was carried out after the policy supposedly showed results. So, the authors must answer: were the participants biased and hence the study biased due to these interventions already in effect? Last but not least, the manuscript requires a thorough grammar check as it has many errors. I would recommend that the authors either hire an editor or get the manuscript edited by someone whose proficiency is at a native speaker level. When these objections are addressed, I am willing to reconsider the manuscript.  
